# Pulsed Vacuum Drying of Persimmon Slices: Drying Kinetics, Physicochemical Properties, Microstructure and Antioxidant Capacity

**DOI:** 10.3390/plants11192500

**Published:** 2022-09-23

**Authors:** Kai-Wen Yang, Dong Wang, Sriram K. Vidyarthi, Suo-Bin Li, Zi-Liang Liu, Hui Wang, Xian-Jun Chen, Hong-Wei Xiao

**Affiliations:** 1College of Engineering, China Agricultural University, 17 Qinghua Donglu, Beijing 100083, China; 2College of Mechanical and Electrical Engineering, Shaanxi University of Science and Technology, 6 Xuefuzhong Road, Weiyangdaxueyuan District, Xi’an 710021, China; 3Department of Biological and Agricultural Engineering, University of California, One Shields Avenue, Davis, CA 95616, USA; 4Jiangsu Bolaike Freezing Technology Development Co., Ltd., Changzhou 213115, China

**Keywords:** persimmon slices, pulsed vacuum drying, drying kinetics, scanning electron microscopy (SEM), total phenolic content, antioxidant capacity

## Abstract

In order to explore an alternative drying method to enhance the drying process and quality of persimmon slices, pulsed vacuum drying (PVD) was employed and the effects of different drying temperatures (60, 65, 70, and 75 °C) on drying kinetics, color, rehydration ratio (RR), microstructure, bioactive compounds, and the antioxidant capacity of sliced persimmons were investigated in the current work. Results showed that the rehydration ratio (RR) of the samples under PVD was significantly higher than that of the traditional hot air-dried ones. Compared to the fresh samples, the dried persimmon slices indicated a decrease in the bioactive compounds and antioxidant capacity. The total phenolic content (TPC) of PVD samples at 70 °C was 87.96% higher than that of the hot air-dried persimmon slices at 65 °C. Interestingly, at 70 °C, the soluble tannin content and TPC of the PVD samples reached the maximum values of 6.09 and 6.97 mg GAE/g, respectively. The findings in the current work indicate that PVD is a promising drying method for persimmon slices as it not only enhances the drying process but also the quality attributes.

## 1. Introduction

Persimmon (*Diospyros kaki* Thunb.) is a popular crop in East Asia and has been cultivated in China for more than 3000 years. Currently, it is widely consumed in many countries [1]. Persimmon is popular due to its attractive color and appearance, sweet taste, and unique chewiness. In addition, it is a health-promoting food as it is rich in sugars, dietary fiber, and bioactive compounds, including tannins, phenols, carotenoids, and ascorbic acid, which have the function of clearing heat and moistening the lungs, as well as anti-carcinogenic, anti-inflammatory, and antioxidant effects [2,3]. Usually, persimmon is consumed in fresh or dried forms. Fresh persimmon, with a relatively high moisture content and climacteric property, is very sensitive to microbial spoilage during storage, leading to low postharvest stability and a short shelf life [4,5]. In addition, it is a seasonal fruit with a short harvest period and due to the lack of industrial processing, fresh persimmon cannot be consumed throughout the year [6]. Therefore, effective processing and preservation technologies are urgently needed to overcome these limitations [7].

Drying is one of the most employed methods for agricultural materials processing since it can prevent microbial growth, improve food stability, prolong shelf-life, and facilitate storage and transportation by reducing the moisture content to a certain degree [8]. In China and some other developing countries, natural drying and hot air drying are frequently used for persimmon fruits and many other agricultural products drying [9,10]. Although natural drying is cost-effective, it takes a long time and the drying environment is uncontrollable and susceptible to weather changes. In addition, the products can be moldy on rainy days and are prone to contamination by dust and insects due to exposure to open environments [11]. Normally, hot air drying is a conventional drying technique for fruits and vegetables since it is affordable, and the equipment is simple and easy to operate [12]. However, the nutritional qualities of the dried products can be adversely affected by high temperatures, prolonged drying time, and the continuous presence of oxygen [5]. Jia et al. [13] found serious degradation of bioactive compounds in persimmon chips after hot air drying, and the final product had greater color degradation since the conditions facilitate browning reactions. Therefore, selecting suitable drying technology and appropriate drying conditions are of great importance in the processing of high-quality persimmon products [6].

Pulsed vacuum drying (PVD) is a novel drying technology that has been developed in recent years. Compared with continuous vacuum drying, PVD is a good alternative since it can minimize energy consumption, as well as maintain product qualities [14]. When the material is kept in the drying chamber under constant pressure for a certain time period, the air layer in direct contact with the surface of the materials may become saturated with water. The successive change in vacuum and atmospheric pressure in the drying chamber can disturb this air layer, leading to the removal of surface moisture more effectively. Meanwhile, the relatively long vacuum duration can inhibit oxidation reaction and reduce the deterioration of product quality [15]. PVD has already been successfully used for processing wolfberry [15], Rhizoma Dioscoreae slices [14], lemon slices [16], Chinese ginger slices [17], kiwifruit slices [18], and blueberry [19,20]. Xie et al. [15] reported that the drying time cost by PVD is dramatically reduced by 81.08% compared to the traditional drying methods with a coal-fired drying room at 70 °C, which indicates that PVD significantly shortened the drying time of wolfberry under appropriate conditions compared with hot air drying. Liu et al. [20] studied the blueberry samples treated by PVD under the pulsed vacuum ratio of 15:4 min at 60, 65, 70, and 75 °C, compared with that of the hot air-dried samples, and found that blueberries obtained from PVD had a higher antioxidant capacity than hot air-dried blueberries due to the lower degradation of bioactive substances and cell wall structures. Wang et al. [16] reported that the color of lemon slices after PVD is better than the samples dried in the closed-type solar dryer, which probably attributes to the fact that the pulsed vacuum environment creates an oxygen-deficient condition and reduces the occurrence of an adverse biochemical reaction. However, to the best of our knowledge, the effect of PVD on the physicochemical properties of persimmon slices after the drying process has not been reported. Persimmon is a kind of fruit with heat-sensitive and easy-browning properties and lacks a suitable processing method at present. Therefore, PVD was applied to persimmon drying in the current study in order to investigate the feasibility of this technology.

As agricultural products are complex in nature, macro- and microstructures are important factors to determine the quality and stability of dried products. An appropriate understanding of the microstructure of food products and their changes under different drying conditions can be essential to understand and accurately describe the food properties [19,21]. Scanning electron microscopy (SEM) has been widely applied for analyzing structural changes after drying. In this study, SEM was used to observe the cell structure of persimmon slices. Various studies have shown that drying treatments can modify the physical and chemical properties of products. During thermal treatment, the structural barriers were destroyed, and the phenolic compounds bound to the cellular structure were released from the matrix, which caused an increase in the bioavailability of nutrients [22,23]. The astringency of persimmon fruit is due to its high soluble tannin content. As a kind of phenolic compound, the soluble tannin content in dried persimmon is a necessary index to evaluate product qualities [1].

Therefore, the purpose of this study was to investigate the effects of PVD under different drying temperatures (60, 65, 70, and 75 °C) with a constant pulsed vacuum ratio (vacuum pressure duration versus atmospheric pressure duration) of 12:4 min on the quality of dried persimmon slices, and to compare the drying kinetics, color parameters (*L**, *a**, *b**, Δ*E*), rehydration ratio, microstructure, bioactive compounds (soluble tannin content, total phenolic content), and antioxidant activity of persimmon slices with those treated by hot air drying (HAD).

## 2. Materials and Methods

The fresh persimmon fruits “Dayaoshan brittle persimmon” were harvested from a plantation at Dayaoshan (Guilin, Guangxi province, China). The fruits were then stored at a temperature of 4 ± 1 °C before processing. For experimentation, the persimmon fruits were washed, pedicles were removed, and fruits were sliced into pieces with a thickness of 4 mm. Slices with almost the same size in the middle of the fruits were selected. The initial moisture content of the fresh samples was 80.5 ± 1.2% (wet basis) based on oven drying at 105 °C.

### 2.1. Drying Methods

#### 2.1.1. Pulsed Vacuum Drying (PVD)

Pulsed vacuum drying experiments were conducted using the PVD equipment, which is situated in the College of Engineering of China Agricultural University, Beijing, China. The schematic diagram of the PVD experiment is shown in Figure 1. The PVD equipment mainly consists of heating, vacuum, and cooling units and control systems, which have been described by Xie et al. [15] in detail.

During the drying process, energy is provided for material dehydration through heat conduction. On the other hand, the pressure in the drying chamber is constantly changed to make it pulse and circulate between atmospheric pressure and vacuum until the materials reach the target moisture content. The change in pressure in the drying chamber during PVD is shown in Figure 2. Firstly, the pressure in the drying chamber changes from atmospheric pressure to vacuum, which is maintained for a certain period, then it is restored to atmospheric pressure and remains as such for a while. This cycle is repeated until the end of drying. The minimum pressure level that the system can generate is 8.0 kPa and the time taken for the system to reach this minimum pressure from atmospheric pressure is approximately 40 s [17].

The pulsed vacuum ratio (vacuum pressure duration versus atmospheric pressure duration) was used in this analysis for descriptive purposes. For example, the pulsed vacuum ratio 12:4 means the vacuum pressure duration and atmospheric pressure duration are 12 and 4 min, respectively. According to a preliminary study, the PVD experiment was divided into 4 groups at different drying temperatures of 60, 65, 70, and 75 °C, respectively, under the same pulsed vacuum ratio of 12:4. The experiments continued until the moisture content of the samples decreased to 15.0 ± 3.0% (wet basis). Each drying treatment was carried out in triplicate.

#### 2.1.2. Hot Air Drying (HAD)

The hot air-drying experiment was conducted using a hot air dryer, which is situated at the College of Engineering of China Agricultural University, Beijing, China. To assess the feasibility of PVD technology in drying persimmon slices, the HAD experiment was carried out at 65 °C with an air velocity of 3 m/s until reaching an equilibrium moisture content of 15.0 ± 3% (wet basis). The drying treatment was repeated three times.

### 2.2. Drying Kinetics

#### 2.2.1. Drying Characteristics

The moisture ratio (*MR*) of persimmon slices was calculated based on Equation (1) [24]:(1)$$MR=\frac{M_{t}-M_{e}}{M_{0}-M_{e}}$$
where *M_t_* is the moisture content at *t* time of drying, kg/kg; *M*_0_ is the initial moisture content, kg/kg; and *M*_e_ is the equilibrium moisture content, kg/kg. All moisture contents were expressed on a dry basis.

The drying rate (*DR*) was calculated based on Equation (2) [25]:(2)$$DR=\frac{M_{t1}-M_{t2}}{t_{2}-t_{1}}$$
where *t*_1_ and *t*_2_ are the specific times during drying, h; and *M_t_*_1_ and *M_t_*_2_ are the moisture contents of persimmon slices at the specific drying times *t*_1_ and *t*_2_, respectively, g/g DW (dry weight).

#### 2.2.2. Kinetic Modeling of Drying Process

The Weibull model seen Equation (3) was used to fit the drying curves of persimmon slices according to Doymaz et al. [9].
(3)$$MR=\exp\left[-{\left(\frac{t}{\alpha }\right)}^{\beta }\right]$$
where *α* is the scale parameter of the Weibull model, *β* is the shape parameter of the model, and *t* is the drying time, min.

The coefficient of determination (*R*^2^), the reduced chi-square parameter (*χ*^2^), and the residual sum squares (*RSS*) between the predicted and experimental values were used to determine the fitting quality of the model to the experimental data, which were calculated by the expression Equations (4)–(6) [17]:(4)$$R^{2}=1-\frac{{\sum_{i=1}^{N}\left(MR_{pre,i}-MR_{exp,i}\right)}^{2}}{{\sum_{i=1}^{N}\left(\bar{MR_{pre,i}}-MR_{exp,i}\right)}^{2}}$$
(5)$$\chi^{2}=\frac{{\sum_{i=1}^{N}\left(MR_{pre,i}-MR_{exp,i}\right)}^{2}}{N-z}$$
(6)$$RSS={\sum_{i=1}^{N}\left(MR_{pre,i}-MR_{exp,i}\right)}^{2}$$
where *MR_pre,i_* and *MR_exp,i_* are experimental and predicted values, respectively; *N* is the number of experiences; and *z* is the number of constants.

### 2.3. Color Measurements

The color of persimmons under different drying treatments was measured by a colorimeter (Lab Scan XE, s/n: LX 18423) and expressed by the CIE *L***a***b** color system. Before measurement, the black and white standard plates were used to correct the color. The lightness of color *L**, red/green value *a**, and blue/yellow value *b** of persimmon slices were determined in sextuplicate for each group. The calculation formula of color difference (Δ*E*) is as follows [26]:(7)$$\Delta E=\sqrt{{\left(L_{0}{}^{*}-L^{*}\right)}^{2}+{\left(a_{0}{}^{*}-a^{*}\right)}^{2}+{\left(b_{0}{}^{*}-b^{*}\right)}^{2}}$$
where *L*_0_*, *a*_0_*, and *b*_0_* are the color parameters of the raw sample; and *L**, *a**, and *b** are the color parameters of the dried samples. The measurements were conducted in sextuplicate.

### 2.4. Rehydration

The rehydration process of dried samples follows the method described by Wang et al. [16] with some modifications. One gram of dried persimmon slices under different treatments were put into a glass beaker with 100 mL distilled water in a constant-temperature water bath at 40 °C for a period of time. Then the rehydrated samples were taken out from water and blotted with a dry paper towel to remove excess water on the surface. After weighing, the samples were put back into the same soaking beaker immediately. This process was repeated until a constant weight was achieved. All experiments were measured in triplicate, and the rehydration ratio (*RR*) of each sample was calculated based on the Formula (8):(8)$$RR=\frac{m_{t}}{m_{0}}$$
where *m_t_* is the weight of persimmon slices after rehydration, g; and *m*_0_ is the weight of persimmon slices before rehydration, g.

### 2.5. Scanning Electron Microscopy (SEM) Observations

The surface microstructure of persimmon slices under different drying conditions was observed by scanning electron microscope (JEOL, SU3500, Hitachi, Japan). Samples with a size of 5 × 5 mm were cut with a scalpel, fixed on the observation table with conductive adhesive, sprayed with gold for 30 s, and observed under the condition of a loading voltage of 15 kV.

### 2.6. Methods of Persimmon Extraction

Approximately 1.0 g of dried persimmon slices were placed in a mortar and ground with 20 mL of 70% methanol. They were transferred into 50 mL centrifuge tubes and extracted with ultrasound equipment for 30 min. Centrifugation was performed at 8000 rpm and 4 °C for 15 min. The supernatant was used to determine the tannin content, total phenolic content, and antioxidant activity.

### 2.7. Soluble Sannin Content

The soluble tannin content of dried persimmon samples was determined according to the method described by Zhao et al. [4] with some modifications. The Folin–Denis method was used to determine tannin content. For determination, 0.2 mL of supernatant was taken and 0.5 mL of Folin–Ciocalteu was added. After 3 min, 1 mL of 20% Na_2_CO_3_ solution was added and shaken well. After 1 h at room temperature (25 °C), the solution was diluted to 10 mL with distilled water and the absorbance of the mixture was measured by spectrophotometer (Beijing General Analysis Instrument Co., LTD., Beijing, China) at 725 nm.

The results were expressed as mg gallic acid equivalents (GAE)/g on a dry weight basis. Gallic acid was used as the standard, and the results were calculated on the basis of a calibration curve and expressed as a gallic acid equivalent (GAE) in mg/g DW.

### 2.8. Total Phenolic Content (TPC)

The determination of the total phenolic content of samples was slightly modified according to the method of Liu et al. [19]. A volume of 0.2 mL of supernatant was accurately absorbed into a 10 mL colorimetric tube, and 0.8 mL of Folin–Ciocalteu reagent and 5 mL of distilled water were added and mixed well. After 10 min, 2 mL of 7.5% Na_2_CO_3_ solution was added and mixed well, and the mixture was kept in the dark at room temperature for 60 min. The absorbance value was measured at 765 nm to calculate the total phenolic content. Three replicates were performed for each sample.

### 2.9. Antioxidant Activity

The DPPH (2,2-diphenyl-1-pyridyl hydrazide) radical scavenging activity and FRAP (ferric reducing antioxidant power) of dried persimmon slices were measured according to the method described by González et al. [26]. Based on dry weight, the results were expressed as μmol Trolox equivalent (TE)/g dry matter.

### 2.10. Statistical Analysis

Each group was conducted in triplicate and the average values were used for plotting. The experimental data were analyzed using a one-way analysis of variance (ANOVA) by SPSS statistical software (Version 22.0, SPSS Inc., Chicago, IL, USA). The Duncan test was used to evaluate the statistical significance of differences at a 5% probability level (*p* < 0.05). The statistical analysis of drying experiments for model fitting was performed with Matlab software (Version 8.5, MathWorks, Natick, MA, USA).

## 3. Results and Discussion

### 3.1. Drying Characteristics and Kinetic Modeling

The drying curves and drying rate curves of persimmon slices under different PVD (12:4 min) conditions are shown in Figure 3 to explore the effects of drying temperatures on the drying characteristics of persimmon slices. From Figure 3a, it can be observed that the moisture content of persimmon slices decreased with the increase in drying time. Under the same pulsed vacuum ratio (12:4), the drying time decreased with the increase in drying temperature. When the drying temperature increased from 60 to 75 °C, the drying time shortened from 412 to 252 min. There was no significant difference in drying time between 60 and 65 °C (*p* < 0.05). This result indicates that increasing the drying temperature can improve the drying speed and shorten the drying time when other factors remain unchanged. This is consistent with the results of Deng et al. [11] and Wang et al. [16], who found that the drying time decreases with an increasing drying temperature.

The relationships between the drying rate and moisture content in the dry basis of persimmon slices at different drying temperatures are shown in Figure 3b. During the initial drying stage, the drying rate increased rapidly. This may be due to the existence of excessive water on the surface of persimmon slices, and the moisture removal rate was very fast. As the drying process proceeds, the evaporation of moisture on the surface of the samples becomes less important, and the process of moisture diffusion within the samples gradually becomes the factor that matters [4]. Obviously, the drying rate increased with the increasing drying temperature (from 60 to 75 °C), and there was no significant difference between 60 and 65 °C (*p* < 0.05). This is consistent with a general perception that a higher drying temperature produces a larger heat and mass transfer driving force, thus increasing the drying rate [24]. These findings are consistent with the results on blueberries and kiwifruit slices [18,19].

The Weibull model was fitted to the drying experimental data of persimmon slices (Table 1). All the coefficients of determination (*R*^2^) of the Weibull model (>0.99) demonstrated goodness of the fitting. The values of *α* and *β* represent scale parameter and shape parameter, respectively. It can be seen from Table 1 that the *α* values decreased with the increasing of drying temperature. The lower the *α* values, the faster the moisture removal rates [27]. The *β* values ranged from 1.294 to 1.453, and lower *β* values indicated a higher drying rate at the beginning [27].

### 3.2. Color Evaluation

The color property is one of the most important sensory evaluation parameters of the dried persimmon slices, which directly impacts the purchasing desire for the product. The color parameters *L**, *a**, *b**, and Δ*E* values of dried persimmon slices under different drying conditions are shown in Table 2.

The *L** values represent the lightness and brightness of the sample surface [11]. It can be seen that the *L** values of fresh persimmon slices are significantly higher than those of PVD and HAD treatments. A wide variation in the levels of lightness was observed in dried persimmon slices. The highest *L** value was observed in the sample dried by HAD at 65 °C (62.39 ± 0.86) and the lowest in the ones dried by PVD at 60 °C (52.02 ± 1.75). The *L** values of HAD-treated samples were higher than those treated by PVD, which is due to the longer processing time of persimmon slices during PVD, resulting in reduced brightness [16].

As shown in Table 2, the *b** values of samples were affected by the drying conditions. Compared with the fresh samples, *b** values increased after the drying process. The highest *b** value was found in persimmon slices treated by HAD at 65 °C (45.52 ± 0.38). These color changes in yellowness (*b** value) are probably associated with browning reactions due to long drying times [12]. There were no significant differences (*p* < 0.05) in *a** values observed among all dried persimmon slices.

The Δ*E* values are often used to evaluate the total color difference between dried and fresh samples. Generally speaking, lower Δ*E* values mean less color variation during heat treatment [23]. It was observed that the Δ*E* values of HAD-treated samples were lower than those of PVD-treated samples, indicating that the color of persimmon slices treated by HAD was closer to the fresh samples with better color quality. The result confirmed that PVD samples underwent greater color degradation, which might be due to the longer drying time of PVD.

### 3.3. Rehydration Ratio

The rehydration results of persimmon slices under different drying conditions are depicted in Figure 4. It can be seen that the rehydration ratio of persimmon slices shows a trend of increasing at first and then decreasing with the enhancement of the drying temperature, and the rehydration ratio reached the maximum of 3.06 at 65 °C treated by PVD. The rehydration capacity can reflect the degree of microstructure damage during drying, such as internal collapse [15]. Lower drying temperatures prolong the drying time, which may cause more serious damage to the material cell wall skeleton, thus reducing the ability of water absorption in the rehydration process. Higher drying temperatures can lead to the collapse and hardening of material cells, which may reduce water permeability during rehydration as the microstructure of the product determines its macroscopic properties [28]. Therefore, a suitable drying temperature can make the rehydration properties of the material achieve the best results. The rehydration ratio of 2.38 of the samples treated by HAD at 65 °C was lower than that of PVD at all temperatures. This may be due to the fact that vacuum conditions cause greater internal stress in the material and the formation of lots of porous structures on the surface and inside the material [24], which makes it easier for water to enter the material during rehydration.

### 3.4. Scanning Electron Microscopy (SEM) Evaluation

In order to explain the differences in the rehydration properties of dried persimmon slices at the microscopic level, the microstructure of dried persimmon slices under different drying methods and temperatures was observed with a scanning electron microscope (Figure 5). The microstructure of persimmon slices was significantly different after drying at different temperatures. It could be seen that the cells of dried persimmon slices were more disordered at 60 °C, and the cytoskeleton collapsed with the loose structure. At 65 °C and treated by PVD, the cell pores arranged neatly, and the structure was more compact, which allowed water to enter the dried persimmon slices easily, making the rehydration effect better. The cell pores were intact at 70 °C and there was no significant difference in the rehydration effect compared to that at 65 °C. Fewer holes were observed on the sample surface at 75 °C, which reveals that the rehydration performance at this temperature was poor. Wang et al. [16] also discovered a similar phenomenon. Compared with PVD samples treated at different temperatures, there were less pore structures on the persimmon slices treated by HAD at 65 °C, resulting in the lowest rehydration ratio in persimmon slices. These results observed by SEM are consistent with the analysis of the rehydration properties of samples, confirming the hypothesis of the reasons for the differences in the rehydration performance of dried persimmon slices. The findings from Figure 5 indicated that significantly prolonged exposure to the thermal environment or higher temperature may cause cells and tissues to collapse, resulting in worse rehydration capacity [20].

### 3.5. Soluble Tannin Content

Tannins are one of the phenolic substances that attribute astringent properties to the substance. Some kinds of persimmon taste astringent due to the presence of soluble tannins, which react with salivary proteases, leading to their complexation, and thus giving rise to an astringent taste in the mouth [29]. Therefore, reducing the soluble tannin content and enhancing the taste of persimmon is of great interest to the industry. In order to explore the effect of drying thermal treatment on the tannin content of persimmon, the soluble tannin content of persimmon slices dried under different conditions is shown in Figure 6. Interestingly, the tannin content of dried persimmon slices generally increased first and then decreased as the drying temperature of PVD increased from 60 to 75 °C, and the highest tannin content of 6.09 (mg GAE/g) was discovered at 70 °C. This result is different from Zhao et al. [4] who found that the soluble tannin content decreased significantly with increasing drying temperatures (*p* < 0.05). This may be due to the longer drying time at lower drying temperatures, resulting in more thermal degradation of tannins. The temperature is higher at 70 °C, and the drying time is shortened accordingly, which makes tannins less likely to be degraded. However, the drying temperature 75 °C is so high that it might lead to the thermal degradation of tannins. Another reason might be the persimmon’s special response to temperature changes. It has been discovered from previous studies that the astringency of persimmon could recur when treated at temperatures above 65 °C [30], but the mechanism is unclear, and it may also be associated with a sudden increase in tannin content. Compared with dried samples at different temperatures, the tannin content of fresh persimmon slices was the highest. Thermal processing, such as PVD and HAD, may be an effective method to reduce the tannin content and astringency of persimmon [4].

### 3.6. Total Phenolic Content (TPC)

The total phenolic content (TPC) of dried persimmon slices under different drying conditions is shown in Figure 7. It can be found that the TPC in dried persimmon slices was significantly lower than that in the fresh samples. In addition, the trend of TPC under different temperatures was consistent with the trend of tannin contents. Considering tannins as a major phenolic substance in persimmon, the change in their contents could be used to show the trend of total phenolic substances [4,31]. From Figure 7, it was also found that the TPC of persimmon slices declined gradually when the drying temperature of PVD increased from 60 to 65 °C, while the TPC reached its maximum value of 6.97 (mg GAE/g) at 70 °C. This may be due to the fact that phenolic compounds are thermally sensitive, and lower drying temperatures caused longer drying times, leading to the degradation of phenolic compounds [32]. Although a higher drying temperature can accelerate the degradation of phenolic substances, it can shorten the drying time and counteract the adverse effects of high temperatures, resulting in a higher retention rate of phenolic compounds. Therefore, in order to preserve the highest content of bioactive compounds during the drying process, the appropriate temperature is particularly important.

The TPC of persimmon slices treated by HAD at 65 °C was lower than that of PVD. Kayacan et al. [32] and Carmona et al. [33] also observed a similar phenomenon. Most likely, irreversible oxidation of phenolic compounds occurred when persimmon slices were exposed to oxygen all the time during HAD.

### 3.7. Antioxidant Capacity

DPPH and FRAP methods were used to express the antioxidant capacity of persimmon slices. The effects of different drying conditions on the antioxidant activity of persimmon slices are shown in Figure 7. The changing trend of DPPH and FRAP were basically the same. When PVD temperature increased to 70 °C, DPPH and FRAP reached their maximum values after drying, which were 54.56 and 66.22 μmol Trolox/g d.m., respectively. While the PVD temperature increased to 75 °C, the values of DPPH and FRAP decreased to the minimum, which were 29.61 and 23.89 μmol Trolox/g d.m., respectively. Therefore, a proper drying temperature is very important to preserve the antioxidant capacity of samples during drying.

It is not difficult to discover the high correlation between antioxidant capacity and TPC results, which indicates that the TPC of persimmon slices is closely related to the antioxidant capacity. Similar results were also observed by González et al. [34] in “Rojo Brillante” persimmon drying, Ketnawa et al. [35] in non-astringent persimmon fruit processing, and Liu et al. [20] on blueberry drying who reported that significant correlations (*p* < 0.05) were observed between DPPH or FRAP and the total phenolics content. The strong antioxidant capacity of persimmon is derived from tannins [36], one of the phenolic substances, so the changes in TPC can directly alter the antioxidant capacity. The DPPH and FRAP values of PVD under different temperatures were higher than those of HAD, which was consistent with the contents of tannin and TPC. Persimmon slices dried by PVD were exposed to the vacuum environment for a long period of time, so the possibility of phenol oxidation was reduced, and consequently, the antioxidant capacity was correspondingly higher. Fresh samples had the highest antioxidant capacity, while the thermal drying process decreased it. This result may be caused by the degradation of bioactive compounds, such as phenols, and the loss of antioxidant enzyme activity due to thermal degradation [37].

## 4. Conclusions

In current work, in order to explore a feasible drying method for persimmon slices, the effect of PVD under different drying temperatures (60, 65, 70, and 75 °C) on its drying kinetics, microstructure, and quality attributes in terms of color parameters (*L**, *a**, *b**, Δ*E*), rehydration ratio, bioactive compounds, and antioxidant capacity was explored and compared with the samples dried by hot air. In terms of the nutrient retention rate of dried persimmon slices, PVD is superior to HAD. The samples treated by PVD obtained higher tannin contents, TPC, and antioxidant capacity than the HAD ones by reducing the degradation of bioactive substances. Moreover, better rehydration performance of dried persimmon slices was noticed after PVD. Microstructure observation showed that PVD samples had more neat and compact pore structures on the surface, which facilitates moisture transfer to enhance its rehydration performance. The findings in the current work indicate that pulsed vacuum drying is a promising method for persimmon drying as it not only enhances the drying process but also the quality attributes.

## Figures and Tables

**Figure 1 plants-11-02500-f001:**
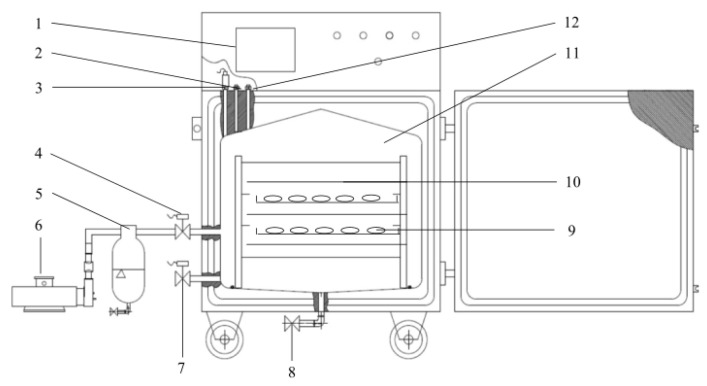
Schematic diagram of pulsed vacuum drying equipment. 1. Touch screen, 2. material temperature sensor, 3. pressure sensor, 4. vacuum valve, 5. condenser, 6. vacuum pump, 7. air solenoid valve, 8. drain solenoid valve, 9. sample, 10. far-infrared radiation heating element, 11. drying chamber, 12. infrared-board temperature sensor.

**Figure 2 plants-11-02500-f002:**
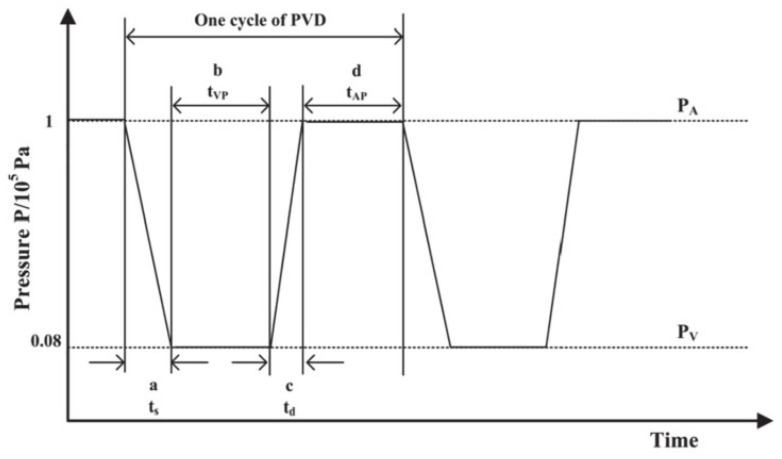
Schematic diagram of pressure-changing kinetics in drying chamber during pulsed vacuum drying (PVD). P_A_: the highest pressure in the drying chamber, P_V_: the lowest pressure in the drying chamber, *t*_AP_: the duration at the highest pressure, *t*_VP_: the duration at the lowest pressure, *t*_s_: the time required during alternation of P_V_, *t*_d_: the time required during alternation of P_A_.

**Figure 3 plants-11-02500-f003:**
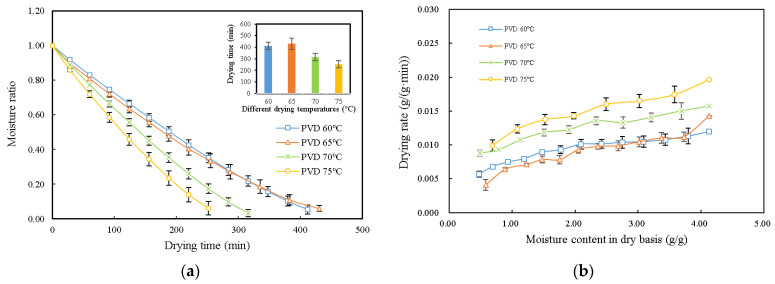
Drying curves and drying rate curves of persimmon slices under different PVD (12:4 min) conditions. PVD: pulsed vacuum drying. (**a**) Moisture ratio versus drying time; (**b**) drying rate versus moisture content in dry basis.

**Figure 4 plants-11-02500-f004:**
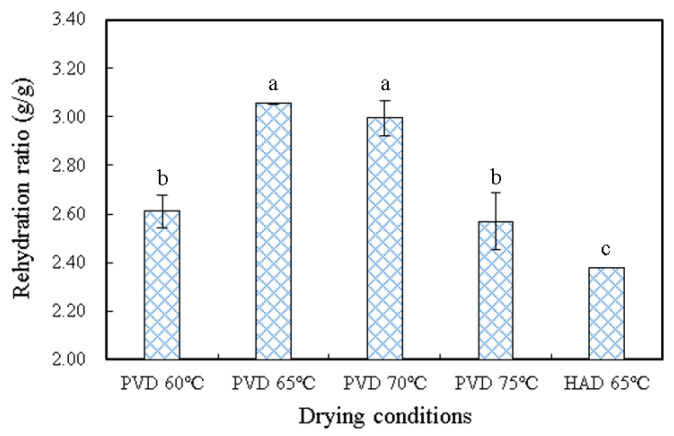
Rehydration ratio of persimmon slices under different drying conditions. PVD: pulsed vacuum drying (12:4 min). HAD: hot air drying. The different letters, a, b, and c, reveal significant differences (*p* < 0.05) according to the Duncan test.

**Figure 5 plants-11-02500-f005:**
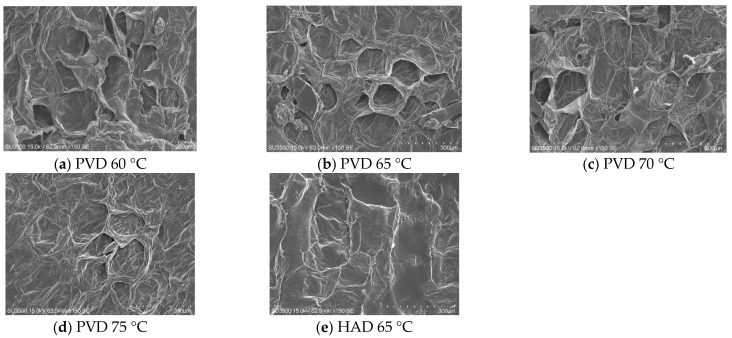
The surface microstructure of dried persimmon slices under different drying conditions. PVD: pulsed vacuum drying (12:4 min). HAD: hot air drying.

**Figure 6 plants-11-02500-f006:**
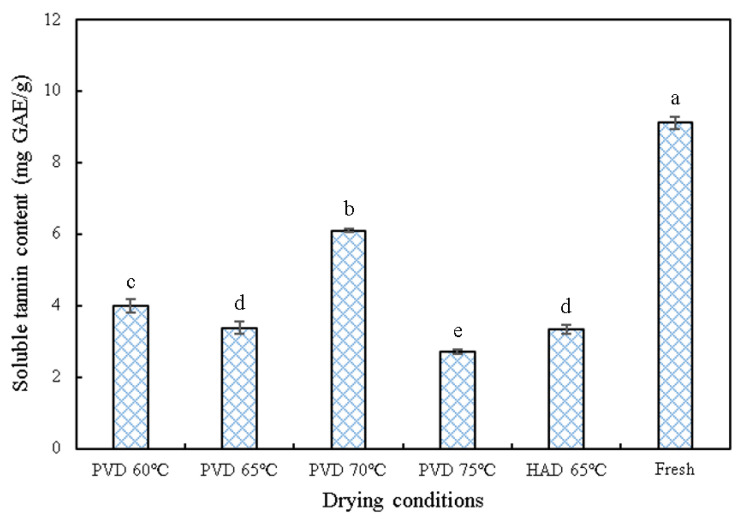
Effects of different drying treatments on soluble tannin content. PVD: pulsed vacuum drying (12:4 min). HAD: hot air drying. The different letters, a, b, c, d, and e, reveal significant differences (*p* < 0.05) according to the Duncan test.

**Figure 7 plants-11-02500-f007:**
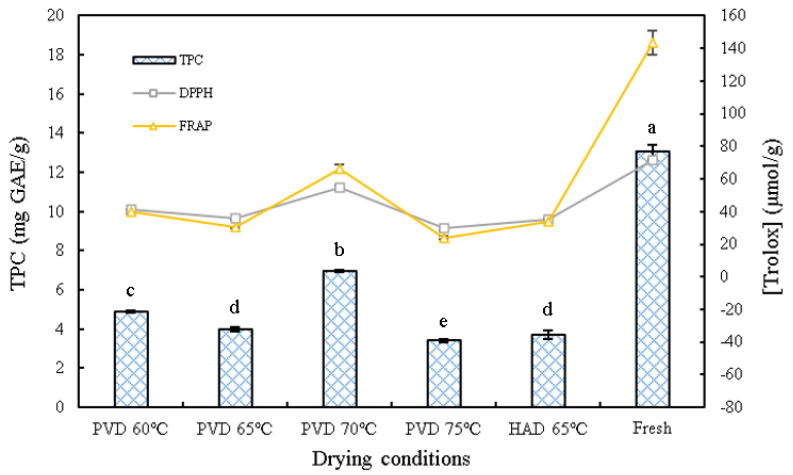
The total phenolic content (TPC), DPPH, and FRAP values of dried persimmon slices at different drying conditions. PVD: pulsed vacuum drying (12:4 min). HAD: hot air drying. The different letters, a, b, c, d, and e, reveal significant differences (*p* < 0.05) according to the Duncan test.

**Table 1 plants-11-02500-t001:** Regression results of Weibull model fitted to moisture ratio changes at different drying temperatures during PVD.

Temperature (°C)	*α* (min)	*β*	*χ*^2^ (×10^−4^)	*RSS*	*R* ^2^
60	233.1	1.422	9.288	0.009	0.993
65	228.0	1.294	7.564	0.007	0.994
70	173.2	1.453	13.012	0.009	0.991
75	141.4	1.387	0.001	0.006	0.993

**Table 2 plants-11-02500-t002:** The color parameters *L**, *a**, *b** and Δ*E* of dried persimmon slices under different drying conditions.

	Fresh 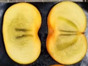	PVD 60 °C 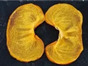	PVD 65 °C 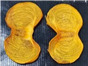	PVD 70 °C 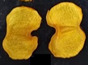	PVD 75 °C 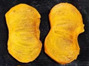	HAD 65 °C 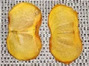
*L**	84.80 ± 0.73 ^a^	52.02 ± 1.75 ^e^	57.46 ± 0.40 ^d^	56.09 ± 0.65 ^d^	53.40 ± 0.96 ^e^	62.39 ± 0.86 ^b^
*a**	7.70 ± 1.09 ^b^	13.48 ± 0.67 ^a^	12.85 ± 0.55 ^a^	13.61 ± 0.16 ^a^	13.71 ± 1.25 ^a^	14.60 ± 0.22 ^a^
*b**	36.49 ± 1.81 ^d^	38.31 ± 0.37 ^c^	41.74 ± 0.83 ^b^	42.27 ± 0.64 ^b^	42.29 ± 1.19 ^b^	45.52 ± 0.38 ^a^
Δ*E*	-	33.34 ± 1.65 ^a^	28.32 ± 0.25 ^b^	29.88 ± 0.62 ^b^	32.52 ± 1.14 ^a^	25.13 ± 0.86 ^c^

Note: The different letters, a, b, c, d, and e, in the same row reveal significant differences (*p* < 0.05) according to the Duncan test.

## Data Availability

Data is contained within the article.

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
