# Peer review of "Pulsed Vacuum Drying of Persimmon Slices: Drying Kinetics, Physicochemical Properties, Microstructure and Antioxidant Capacity"

_plants, 2022, doi:10.3390/plants11192500_

Round 1

Reviewer 1 Report

In this study, the authors evaluated the pulsed vacuum drying method for persimmon slices at different temperatures (60, 65, 70, and 75 °C) by observing the drying kinetics, color, rehydration ratio, micro-structures, bioactive compounds, and antioxidant capacity. Finally, the authors concluded that pulsed vacuum drying improved the drying process as well as the quality attributes. The manuscript is good however, the following points need to be addressed:

1.     1. Abstract should be revised. The end line of the abstract should contain the best results (60, 65, 70, and 75 °C).

22.       Keywords should be arranged alphabetically.

33.       Provide the citation for lines 49-51.

44.       Mention the name of the persimmon variety which was used in the experiment.

55.       Authors claimed that fruit was harvested at Guilin and pulsed vacuum drying/hot air drying was carried out at Beijing (China Agricultural University). According to my knowledge, these two localities are far away (20 hours, if I am not wrong). But the authors didn’t mention the transportation of fruit in the M&M.

66.       Figure 1 showed the schematic diagram of pulsed vacuum drying equipment. The figure caption should also contain the names of parts 1,2,3, etc.

77.       The detail of figure 2 should also be provided with a(ts), b(tvp), etc. labels.

88.       The heading at line 208 is not correct.

99.       The method for total phenolic content and soluble tannin content is the same.

110.   Line 416-417, “probably” should be mentioned only in the discussion. Revise and improve the conclusion.

Author Response

In this study, the authors evaluated the pulsed vacuum drying method for persimmon slices at different temperatures (60, 65, 70, and 75 °C) by observing the drying kinetics, color, rehydration ratio, micro-structures, bioactive compounds, and antioxidant capacity. Finally, the authors concluded that pulsed vacuum drying improved the drying process as well as the quality attributes.

Response: Thanks very much for your feedback.

The manuscript is good however, the following points need to be addressed:

Point 1: Abstract should be revised. The end line of the abstract should contain the best results (60, 65, 70, and 75 °C).

Response 1: Thank you very much for your valuable feedback. In our study, different indicators have different trends with temperatures change. Some indicators perform best at 65°C during PVD, while others perform best at 70°C. Therefore, it couldn’t be generalized to summarize the best results at a particular temperature.

Point 2: Keywords should be arranged alphabetically.

Response 2: Thank you very much for your feedback. We arranged the keywords in the order in which they appear in the article. Referring to the format of previously published articles in this journal, the keywords section is not in alphabetical order.

Point 3: Provide the citation for lines 49-51.

Response 3: Thank you very much for your valuable suggestion. Now we have added the citation for lines 49-51.

The references are as follows:

Doymaz, I. Evaluation of some thin-layer drying models of persimmon slices (Diospyros kaki L.). Energy Conversion and Management 2012, 56, 199-205.

González, C.M.; García, A.L.; Llorca, E.; Hernando, I.; Atienzar, P.; Bermejo, A.; Moraga, G.; Quiles, A. Carotenoids in dehydrated persimmon: Antioxidant activity, structure, and photoluminescence. LWT 2021, 142, 111007.

Point 4: Mention the name of the persimmon variety which was used in the experiment.

Response 4: Thank you very much for your helpful suggestion. Now we have added the name ”Dayaoshan brittle persimmon” of the persimmon variety used in this experiment.

Point 5: Authors claimed that fruit was harvested at Guilin and pulsed vacuum drying/hot air drying was carried out at Beijing (China Agricultural University). According to my knowledge, these two localities are far away (20 hours, if I am not wrong). But the authors didn’t mention the transportation of fruit in the M&M.

Response 5: Thank you very much for your valuable suggestion. Fresh persimmon fruits used in this experiment were picked from Guilin, but we bought them online, and the fruit is vacuum-packed so it has little impact on quality.

Point 6: Figure 1 showed the schematic diagram of pulsed vacuum drying equipment. The figure caption should also contain the names of parts 1,2,3, etc.

Response 6: Thank you very much for your helpful opinions. Now we have added the names of parts 1,2,3, etc. of Figure 1 in the revised maniscript, as follows:

  1. Touch screen, 2. Material temperature sensor, 3. Pressure sensor, 4. Vacuum valve, 5. Condenser, 6. Vacuum pump, 7. Air solenoid valve, 8. Drain solenoid valve, 9. Sample, 10. Far-infrared radiation heating element, 1 1. Drying chamber, 12. Infrared-board temperature sensor.

Point 7: The detail of figure 2 should also be provided with a(ts), b(tvp), etc. labels.

Response 7: Thank you very much for your helpful opinions. Now we have added the labels of a(ts), b(tvp), etc. of Figure 2 in the revised maniscript, as follows:

PA: the highest pressure in drying chamber, PV: the lowest pressure in drying chamber, tAP: the duration at the highest pressure, tVP: the duration at the lowest pressure, ts: the time required during alternation of PV, td: the time required during alternation of PA.

Point 8: The heading at line 208 is not correct.

Response 8: Thank you very much for your valuable opinions. Now we have corrected the heading as follows:

2.8. Total phenolic content (TPC)

Point 9: The method for total phenolic content and soluble tannin content is the same.

Response 9: Thank you very much for your feedback. Tannin is a kind of polyphenol, so the content of both is determined similarly. The amount and concentration of reagents added are different, and the wavelength of the final determination is different. The determination method was based on the method of Zhao et al. and Liu et al. with slight modifications.

The references are as follows:

Zhao, C.C.; Ameer K.; Eun, J.B. Effects of various drying conditions and methods on drying kinetics and retention of bioactive compounds in sliced persimmon. LWT 2021, 143, 111149.

Liu, Z.L.; Xie, L.; Zielinska, M.; Pan, Z.L.; Wang, J.; Deng, L.Z.; Wang, H.; Xiao, H.W. Pulsed vacuum drying enhances drying of blueberry by altering micro-, ultrastructure and water status and distribution. LWT 2021, 142, 111013.

Point 10: Line 416-417, “probably” should be mentioned only in the discussion. Revise and improve the conclusion.

Response 10: Thank you very much for your helpful suggestion. Now we have modified this section as follows:

In current work, in order to explore a feasible drying method for persimmon slices, the effect of PVD under different drying temperatures (60, 65, 70, and 75 °C) on its drying kinetics, microstructure, and quality attributes in terms of color parameters (L*, a*, b*, ΔE), rehydration ratio, bioactive compounds, and antioxidant capacity were explored and compared with the samples dried by hot air. In terms of nutrient retention rate of dried persimmon slices, PVD is superior to HAD. The samples treated by PVD obtained higher tannin content, TPC, and antioxidant capacity than HAD ones by reducing the degradation of bioactive substances. Moreover, better rehydration performance of dried persimmon slices was noticed after PVD. Microstructure observation showed that PVD samples had more neat and compact pore structures on the surface, which facilitates moisture transfer to enhance its rehydration performance. The findings in current work indicate that pulsed vacuum drying is a promising method for persimmon drying as it not only enhances the drying process but also the quality attributes.

Reviewer 2 Report

Within this work, the investigation of an alternative drying method was presented, with an idea to enhance the drying process and quality of persimmon slices. The pulsed vacuum drying was employed and the effects of different drying temperatures (60, 65, 70, and 75 °C) on drying kinetics, color, rehydration ratio, micro-structure, bioactive compounds and antioxidant capacity of sliced persimmons were investigated in current work.

This is an extensive research, with a lot of numerical analysis. 

More information should be presented in chapter 3.1., how the calculation was performed? Which kinetics model were used in calculation? The same comment could be applied to chapters 3.2. and 3.3. No kinetics model were presented, no regression coefficients were shown, but the title of the paper the "kinetics" term was mentioned?

Thematically the work is interesting for the researchers and professionals and the proposed manuscript is relevant to the scope of the journal.

There are no information of variable's influence on the responses? At least an ANOVA calculation should be presented for data shown in Fig. 3 and Fig. 4, and Table 1?

SEM analysis is presented in Fig. 5, but no real discussion of the results was not presented, and it is hard to follow the text? "The cell pores were intact at 70 °C, so there was no significant difference in the rehydration 318 effect with that of 65 °C". Please include the data regarding "cell pores" dimensions in the text.

I found it appropriate for publication in the Plants journal, but only after some modifications and clarification from the Authors.

The title is a clear representation of the manuscript's content. Please have in mind the before mentioned sentence that no kinetics model were presented, no regression coefficients were shown, but the title of the paper the "kinetics" term was mentioned?

The abstract reflects realistically the substance of the work.

The overall organization and structure of the manuscript are appropriate.  The paper is relatively well written and the topic is appropriate for the journal. Please have in mind that more numerically expressed data should be presented throughout the text, rather that presenting generalized explanation of the results.

The aim of the paper is well described and the discussion was well approached, its results and discussion are correlated to the cited literature data. Again, please have in mind that more numerically expressed data should be presented throughout the text, rather that presenting generalized explanation of the results.

The literature review is comprehensive and properly done.

The novelty of the work must be more clearly demonstrated. The novelty of the results could be presented more easily if more numerically expressed data should be presented throughout the text, rather that presenting generalized explanation of the results.

The significance of the Work: Given the large number of analyzed data, this is an interesting study with a possible significant impact in this area. The manuscript could have  a much stronger impact in the area if the comments of the reviewer are followed.

Statistical interpretation of the analytical data must be more properly presented. Kinetics model was not presented! The model should be presented, and the verification of the model should be also performed. 

Other Specific Comments: The work is properly presented in terms of the language. The work presented here is very interesting and well done, it is presented in a compact manner.

The main drawback of the paper i s the extent of novelty, or the main novelty in the present work, compared to the works of other researchers? In my opinion, the authors should put additional effort to demonstrate that the present work gives a substantial contribution in the research area.

Author Response

Within this work, the investigation of an alternative drying method was presented, with an idea to enhance the drying process and quality of persimmon slices. The pulsed vacuum drying was employed and the effects of different drying temperatures (60, 65, 70, and 75 °C) on drying kinetics, color, rehydration ratio, micro-structure, bioactive compounds and antioxidant capacity of sliced persimmons were investigated in current work.

This is an extensive research, with a lot of numerical analysis.

Response: Thanks very much for your feedback.

Point 1: More information should be presented in chapter 3.1., how the calculation was performed? Which kinetics model were used in calculation? The same comment could be applied to chapters 3.2. and 3.3. No kinetics model were presented, no regression coefficients were shown, but the title of the paper the "kinetics" term was mentioned?

Response 1: Thank you very much for your valuable question. Originally, this study did not want to highlight the kinetics part, but focused on the analysis of drying characteristics and quality attributes, but that was not rigorous enough. Now we have added the content of drying kinetics modeling as follows:

Weibull model were fitted to the drying experimental data of persimmon slices (Table 1). All the coefficients of determination (R2) of Weibull model (>0.99) demonstrated goodness of the fitting. The values of α and β represent scale parameter and shape parameter, respectively. It can be seen from Table 1 that the α values decreased with the increasing of drying temperature. The lower α values, the faster moisture removal rates [27]. The β values range from 1.294 to 1.453, and the lower β values indicate the higher drying rate at the beginning [27].

Table 1. Regression results of Weibull model fitted to moisture ratio changes at different drying temperatures during PVD.

Temperature (°C)

α (min)

β

ê­“2 ( × 10-4)

RSS

R2

60

233.1

1.422

9.288

0.009

0.993

65

228.0

1.294

7.564

0.007

0.994

70

173.2

1.453

13.012

0.009

0.991

75

141.4

1.387

0.001

0.006

0.993

Thematically the work is interesting for the researchers and professionals and the proposed manuscript is relevant to the scope of the journal.

Response: Thanks very much for your feedback.

Point 2: There are no information of variable's influence on the responses? At least an ANOVA calculation should be presented for data shown in Fig. 3 and Fig. 4, and Table 1?

Response 2: Thank you very much for your feedback. ANOVA calculation have been performed in Excel and are shown as error lines in Figure 3 and Figure 4. In Table 1, SPSS was used for ANOVA calculation to obtain the significance level, and a, b, c, etc. were labeled in the upper right corner of each data.

Point 3: SEM analysis is presented in Fig. 5, but no real discussion of the results was not presented, and it is hard to follow the text? "The cell pores were intact at 70 °C, so there was no significant difference in the rehydration 318 effect with that of 65 °C". Please include the data regarding "cell pores" dimensions in the text.

Response 3: Thank you very much for your valuable question. Now we have modified this sentense as follows: The cell pores were intact at 70 °C and there was no significant difference in the rehy-dration effect compared to that at 65 °C.

It refers to that in Fig. 4, after PVD at 65℃ and 70℃, ANOVA analyse showed that there was no significant difference in rehydration ratio of persimmon slices.

The results of SEM observation are closely related to the rehydration performance, and some analyses have been clarified in the content section of 3.3. rehydration ratio. Now we have added some contents in the SEM result analysis part as follows:

In order to explain the differences in rehydration properties of dried persimmon slices at the microscopic level, the microstructure of dried persimmon slices under different drying methods and temperatures was observed by scanning electron microscope (Figure 5). The microstructure of persimmon slices was significantly different after drying at different temperatures. It could be seen that the cells of dried persimmon slices were more disordered at 60 °C, and the cytoskeleton collapsed with the loose structure. At 65 °C, the cell pores arranged neatly and the structure was more compact, which allowed water to enter the dried persimmon slices easily, making the rehydration effect better. The cell pores were intact at 70 °C and there was no significant difference in the rehydration effect compared to that at 65 °C. Fewer holes were observed on the sample surface at 75 °C, which reveals that the rehydration performance at this temperature was poor. Wang et al. [15] also discovered the similar phenomenon. Compared with PVD samples treated at different temperatures, there were less pore structures on the persimmon slices treated by HAD at 65 °C, resulting in the lowest rehydration ratio of persimmon slices. These results observed by SEM are consistent with the analysis of the rehydration properties of samples, confirming the hypothesis of the reasons for the differences in rehydration performance of dried persimmon slices. The findings from Figure 5 indicated that significantly prolonged exposure to the thermal environment or higher temperature may cause cells and tissues to collapse, resulting in worse rehydration capacity [19].

I found it appropriate for publication in the Plants journal, but only after some modifications and clarification from the Authors.

Response: Thanks very much for your feedback. Now we have made some modifications and improvement in the manuscript.

Point 4: The title is a clear representation of the manuscript's content. Please have in mind the before mentioned sentence that no kinetics model were presented, no regression coefficients were shown, but the title of the paper the "kinetics" term was mentioned?

Response 4: Thank you very much for your helpful suggestion. Now we have added the content of drying kinetics modeling in the revision of the manuscript. The drying kinetics of PVD at different drying temperatures was modeled by Weibull model with Matlab software.

The abstract reflects realistically the substance of the work.

Response: Thanks very much for your feedback.

Point 5: The overall organization and structure of the manuscript are appropriate.  The paper is relatively well written and the topic is appropriate for the journal. Please have in mind that more numerically expressed data should be presented throughout the text, rather that presenting generalized explanation of the results.

Response 5: Thank you very much for your helpful suggestion. Now we have added some numerically expressed data throughout the text in the revision of the manuscript.

Point 6: The aim of the paper is well described and the discussion was well approached, its results and discussion are correlated to the cited literature data. Again, please have in mind that more numerically expressed data should be presented throughout the text, rather that presenting generalized explanation of the results.

Response 6: Thank you very much for your helpful opinions. Now we have added some numerically expressed data and modified the explanation of the results in the revision of the manuscript.

The literature review is comprehensive and properly done.

Response: Thank you very much for your affirmation.

Point 7: The novelty of the work must be more clearly demonstrated. The novelty of the results could be presented more easily if more numerically expressed data should be presented throughout the text, rather that presenting generalized explanation of the results.

Response 7: Thank you very much for your valuable suggestions. Now we have added some numerically expressed data and made some modifications and improvement in the explanation of the results in the revision of the manuscript.

The significance of the Work: Given the large number of analyzed data, this is an interesting study with a possible significant impact in this area. The manuscript could have  a much stronger impact in the area if the comments of the reviewer are followed.

Response: Thank you very much for your affirmation of our research and your valuable opinions.

Point 8: Statistical interpretation of the analytical data must be more properly presented. Kinetics model was not presented! The model should be presented, and the verification of the model should be also performed.

Response 8: Thank you very much for your valuable suggestion. Now we have added the content of drying kinetics modeling in the revision of the manuscript. The drying kinetics of PVD at different drying temperatures was modeled by Weibull model with Matlab software. We have added the drying kinetics model as follows:

Weibull model Eq. (3) was used to fit the drying curves of persimmon slices according to Doymaz et al. [9].

                                   (3)

Where α is the scale parameter of the Weibull model, β is the shape parameter of the model, t is the drying time, min.

The coefficient of determination (R2), the reduced chi-square parameter (χ2), and residual sum squares (RSS) between the predicted and experimental values were used to determine the fitting quality of the model to the experimental data, which were calculated by the expression Eqs. (4)–(6) [16]:

                             (4)

                                (5)

                                 (6)

where, MRpre,i and MRexp,i are experimental and predicted values, respectively; N is the number of experiences and z is the number of constants.

Weibull model were fitted to the drying experimental data of persimmon slices (Table 1). All the coefficients of determination (R2) of Weibull model (>0.99) demonstrated goodness of the fitting. The values of α and β represent scale parameter and shape parameter, respectively. It can be seen from Table 1 that the α values decreased with the increasing of drying temperature. The lower α values, the faster moisture removal rates [27]. The β values range from 1.294 to 1.453, and the lower β values indicate the higher drying rate at the beginning [27].

Table 1. Regression results of Weibull model fitted to moisture ratio changes at different drying temperatures during PVD.

Temperature (°C)

α (min)

β

ê­“2 ( × 10-4)

RSS

R2

60

233.1

1.422

9.288

0.009

0.993

65

228.0

1.294

7.564

0.007

0.994

70

173.2

1.453

13.012

0.009

0.991

75

141.4

1.387

0.001

0.006

0.993

Other Specific Comments: The work is properly presented in terms of the language. The work presented here is very interesting and well done, it is presented in a compact manner.

Response: Thank you very much for your affirmation of our research.

Point 9: The main drawback of the paper i s the extent of novelty, or the main novelty in the present work, compared to the works of other researchers? In my opinion, the authors should put additional effort to demonstrate that the present work gives a substantial contribution in the research area.

Response 9: Thank you very much for your valuable opinions. The purpose of this work is to explore an alternative drying method to enhance the drying process and quality of persimmon slices, pulsed vacuum drying (PVD) was employed in current work. The drying characteristics of persimmon slices during PVD drying at different temperatures, the quality of persimmon slices after drying and the retention of nutrients were studied. SEM was used to analyze the macro rehydration performance from the micro level. This study combined macroscopic phenomena with microscopic structures to characterize the quality of dried persimmon slices in multiple dimensions. The findings in current work indicate that pulsed vacuum drying is a promising method for persimmon drying as it not only enhances the drying process but also the quality attributes. The results of this study will further deepen or change people's understanding of the quality change of persimmon slices after drying, and help to select appropriate drying parameters to obtain more satisfactory products.

Reviewer 3 Report

Pulsed vacuum drying of persimmon slices: drying kinetics, physicochemical properties, microstructure and antioxidant capacity

The manuscript is about the effect of vacuum pressure drying on drying kinetics and drying curve. The proposed technique is interesting and useful for energy efficient and sustainable drying of persimmon. The research findings are well explained. However, there are some errors that need to be addressed. I recommend the article for publication after addressing the following comments.

Abstract

Line 24: “samples was increased by 35.23% than that of the hot 24 air-dried”, word increased is appropriate when it is measured before and after treatment for the same sample. Authors can write “The total phenolic content (TPC) of PVD samples was 35.23% higher than that of the hot air-dried persimmon slices”.

Introduction

Line 37-40: Provide average quantitative data (mg/g).

Line 47-63: Principle of natural drying, its effects and causes are well known. Authors shall reduce the paragraph.

Line 64-86: Authors have written about the mechanism of PVD. It would be more informative if provided with numerical values such as effect pressure and temperature combination on rate of water evaporation. Authors also explained about the similar research by other scientist, kindly provide numerical data of their research as well.

Materials and Methods

Pulsed vacuum drying (PVD): What was the vacuum pressure? Give value for all the ratios.

What was the basis for selecting drying temperature 60 °C, 65 °C, 70 °C, and 75 °C? Since it is at vacuum pressure, drying at lower temperature may also give good results with lesser energy consumption.

Better to explain different treatments in table form.

Results and Discussion

Line 232 233: Was there a significant difference between any other pair of temperatures?

The drying curve and drying rate curve are not compared with the hot air drying method. What was the time in HAD at 65 & 70°C? Explain all the details and modify curves accordingly.

Was there a significant difference in the colour of PVD and HAD? Explain with reason.

Figure 4: HAD 70°C is missing in all quality analyses.

Conclusion: Authors have written the summary of results in the conclusion. Authors should write about how this technique is beneficial over HAD and its industrial applications.

English grammar needs to be improved throughout the manuscript.

Author Response

Response to Reviewer 3 Comments

The manuscript is about the effect of vacuum pressure drying on drying kinetics and drying curve. The proposed technique is interesting and useful for energy efficient and sustainable drying of persimmon. The research findings are well explained. However, there are some errors that need to be addressed. I recommend the article for publication after addressing the following comments.

Response: Thank you very much for your feedback.

Point 1: Line 24: “samples was increased by 35.23% than that of the hot air-dried”, word increased is appropriate when it is measured before and after treatment for the same sample. Authors can write “The total phenolic content (TPC) of PVD samples was 35.23% higher than that of the hot air-dried persimmon slices”.

Response 1: Thank you very much for your helpful suggestion. Now we have modified this sentence as follows:

The total phenolic content (TPC) of PVD samples at 70 °C was 87.96% higher than that of the hot air-dried persimmon slices at 65°C.

Point 2: Line 37-40: Provide average quantitative data (mg/g).

Response 2: Thank you very much for your suggestion. Most of the articles just stated the nutrients that persimmon fruits rich in (for example: Persimmon fruit is rich in health-benefits components such as phenolic compounds, carotenoids, vitamins, and dietary fiber [1].), but did not specify the average quantitative data (mg/g). We only found a relevant statement as follows: Persimmon is rich in some nutrients such as vitamin C (70 mg/100 g of pulp), vitamin A (65 mg/100 g of pulp), calcium (9 mg/100 g of pulp), and iron (0.2 mg/100 g of pulp) [2].

[1] Zhou, M.; Chen, J.X.; Bi, J.F.; Li, X.; Xin, G. The roles of soluble poly and insoluble tannin in the enzymatic browning during storage of dried persimmon. Food Chemistry 2022, 366, 130632.

[2] Nicoleti, J.F.; Jr. V.S.; Romero, J.T.; Telis, V.R.N. Influence of drying conditions on ascorbic acid during convective drying of whole persimmons. Drying Technology 25 (4-6), 891-899.

Point 3: Line 47-63: Principle of natural drying, its effects and causes are well known. Authors shall reduce the paragraph.

Response 3: Thank you very much for your helpful suggestion. In the range of Line 47-63, there are only two sentences to describe the advantages and disadvantages of natural drying, the other parts are the necessity of drying and the advantages and disadvantages of hot air drying technology. Now we have modified this section in the manuscript as follows:

Although natural drying is cost-effective, it takes long time and the drying environment is uncontrollable and susceptible to weather changes. In addition, the products can be moldy in rainy days and are prone to contamination by dust and insects due to exposure to open environment.

Point 4: Line 64-86: Authors have written about the mechanism of PVD. It would be more informative if provided with numerical values such as effect pressure and temperature combination on rate of water evaporation. Authors also explained about the similar research by other scientist, kindly provide numerical data of their research as well.

Response 4: Thank you very much for your helpful opinion. Now we have added some research results of other researchers on PVD in the revision of this manuscript as follows:

Xie et al. [14] reported that the drying time cost by PVD is dramatically reduced by 81.08% compared to the traditional drying methods with coal-fired drying room at 70 °C, which indicates that PVD significantly shortened the drying time of wolfberry under appropriate conditions compared with hot air drying. Liu et al. [19] studied the blueberry samples treated by PVD under the pulsed vacuum ratio of 15 min: 4 min at 60, 65, 70, and 75°C, compared with that of the hot air dried samples and found that blueberries obtained from PVD had higher antioxidant capacity than hot air-dried blueberries due to lower degradation of bioactive substances and cell wall structures.

Point 5: Pulsed vacuum drying (PVD): What was the vacuum pressure? Give value for all the ratios.

Response 5: Thank you very much for your valuable question. The minimum pressure level that the system can generate is 8.0 kPa and the time taken for the system to reach this minimum pressure from atmospheric pressure is approximately 40 s. According to the preliminary experiment, we only selected the most appropriate pulsation ratio of 12:4 to conduct the experiment, studying the influence of temperature change on the drying results. The corresponding modifications are made in the manuscript.

Point 6: What was the basis for selecting drying temperature 60 °C, 65 °C, 70 °C, and 75 °C? Since it is at vacuum pressure, drying at lower temperature may also give good results with lesser energy consumption.

Response 6: Thank you very much for your helpful question. According to the results of our preliminary experiments and previous PVD experiments conducted by other investigators in our study group, the temperatures of 60, 65, 70, and 75 °C has been chosen.

In the PVD (pulsed vacuum drying) process, vacuum pressure and atmospheric pressure are alternated, the pressure in the drying chamber changes from atmospheric pressure to vacuum, and maintains for a certain period, then it restores to atmospheric pressure, and remains for a while. This cycle is repeated until the end of drying. Since it doesn’t maintain vacuum pressure all the time, it is more economical. What’s more, the relatively higher temperature can improve the drying rate, reduce the drying time to a certain extent, and lesser energy consumption.

Point 7: Line 232 233: Was there a significant difference between any other pair of temperatures?

Response 7: Thank you very much for your helpful question. We used SPSS software to analyze the data, and there was a significant difference between any other pair of temperatures.

Point 8: The drying curve and drying rate curve are not compared with the hot air drying method. What was the time in HAD at 65 & 70°C? Explain all the details and modify curves accordingly.   

Response 8: Thank you very much for your valuable opinions. In this study, we did not compare the drying time of persimmon slices in PVD and HAD, but focused on comparing the effects of PVD and HAD on the physical properties and retention of nutritional quality of persimmon slices.

Point 9: Was there a significant difference in the colour of PVD and HAD? Explain with reason.

Response 9: Thank you very much for your valuable question. We used SPSS software to analyze the data and found that there was significant differences between different color parameter values. The specific differences between data have been marked with a, b, c, etc. in the upper right corner of the data in Table 2, and are explained in the analysis.

Point 10: Figure 4: HAD 70°C is missing in all quality analyses.

Response 10: Thank you very much for your valuable feedback. During the experiment, we only use the temperature of 65 °C to study the effect of HAD on persimmon slices and compared to PVD samples. The relevant misstatements have been corrected in the manuscript.

Point 11: Authors have written the summary of results in the conclusion. Authors should write about how this technique is beneficial over HAD and its industrial applications.

Response 11: Thank you very much for your valuable suggestion. Now we have modified this section in the manuscript as follows:

In current work, in order to explore a feasible drying method for persimmon slices, the effect of PVD under different drying temperatures (60, 65, 70, and 75 °C) on its drying kinetics, microstructure, and quality attributes in terms of color parameters (L*, a*, b*, ΔE), rehydration ratio, bioactive compounds, and antioxidant capacity were explored and compared with the samples dried by hot air. In terms of nutrient retention rate of dried persimmon slices, PVD is superior to HAD. The samples treated by PVD obtained higher tannin content, TPC, and antioxidant capacity than HAD ones by reducing the degradation of bioactive substances. Moreover, better rehydration performance of dried persimmon slices was noticed after PVD. Microstructure observation showed that PVD samples had more neat and compact pore structures on the surface, which facilitates moisture transfer to enhance its rehydration performance. The findings in current work indicate that pulsed vacuum drying is a promising method for persimmon drying as it not only enhances the drying process but also the quality attributes.

Point 12: English grammar needs to be improved throughout the manuscript.

Response 12: Thank you very much for your helpful suggestion. Now we have checked the English structure and grammar by a native English speaker in this manuscript. With the help of Dr. Vidyarthi S.K. (University of California, Davis), the quality of manuscript has been improved throughout.

Round 2

Reviewer 2 Report

The Authors managed to improve the quality of the Manuscript according to the Reviewer's comments.

I suggest the Editor to accept the Manuscript in the presented form for possible publication in the Plants journal.

Author Response

The Authors managed to improve the quality of the Manuscript according to the Reviewer's comments.

I suggest the Editor to accept the Manuscript in the presented form for possible publication in the Plants journal.

Response: Thank you very much for your affirmation of our work.

Reviewer 3 Report

Thank you for sending the revised manuscript. The authors have made a sincere effort to improve the manuscript and revised manuscript according to suggestions. I am satisfied with the author's revision except for the quality parameters of a persimmon fruit dried using HAD at 70°C. The authors mentioned that they use only HAD at 65°C for quality analysis but haven't mentioned the reason. The selection should not be based on random personal will but should have a scientific reason. I recommend the manuscript for publication after minor revision in this regard.

Author Response

Response: Thank you very much for your affirmation and valuable question. It has been reported from previous studies that the astringency of persimmon could recur when treated at temperatures above 65 °C [1]. Therefore, temperatures above 65 °C may have certain adverse effects on the qualities of persimmon slices, such as the soluble tannin content. According to the results of the preliminary experiment, the qualities measured under the condition of HAD at 65 °C was the best. Therefore, the HAD experiment was carried out at 65 °C to compare with PVD and assess the feasibility of PVD technology in drying persimmon slices. Now we have modified this section as follows:

Hot air drying experiment was conducted using a hot-air dryer, which is situated in the College of Engineering of China Agricultural University, Beijing, China. To assess the feasibility of PVD technology in drying persimmon slices, HAD experiment was carried out at 65 °C with the air velocity of 3 m/s until an equilibrium moisture content of 15.0 ± 3% (wet basis). The drying treatment was repeated for three times.

[1] Arie, R.B.; Sonego, L. Temperate affects astringency removal and recurrence in persimmon. J. Food Sci. 1993, 58(6), 1397-1400.
